# Duality and hidden equilibrium in transport models

**Rouven Frassek[1][*], Cristian Giardinà[2][†] and Jorge Kurchan[1][‡]**

**1** Laboratoire de Physique de l'École Normale Supérieure, ENS, Université PSL, CNRS,
Sorbonne Université, Université de Paris, 75005 Paris, France
**2** University of Modena and Reggio Emilia, FIM, Via G. Campi 213/b, 41125 Modena, Italy

[*] rouven.frassek@phys.ens.fr, [†] cristian.giardina@unimore.it, [‡] kurchan.jorge@gmail.com

## Abstract

A large family of diffusive models of transport that have been considered in the past
years admit a transformation into the same model in contact with an equilibrium bath.
This mapping holds at the full dynamical level, and is independent of dimension or topol-
ogy. It provides a good opportunity to discuss questions of time reversal in out of equi-
librium contexts. In particular, thanks to the mapping one may define the free energy
in the non-equilibrium states very naturally as the (usual) free energy of the mapped
system.


# 1 Introduction

The purpose of this paper is to show that a large class of models of transport that have been studied by the mathematical physics community in the past years are, in fact, hidden equilibrium models. Stochastic energy transport models were already introduced by Kac [1] for a fully connected graph. On a one dimensional lattice with leads at the ends connected to two reservoirs, two extensively studied models are: the Kipnis-Marchioro-Presutti (KMP) model [2] with thermal baths, where energies are randomly redistributed among nearest-neighbor sites and the symmetric exclusion process (SEP) [3, 4] with particle baths, where particles jump stochastically between neighboring sites.

In a remarkable series of papers (see [5] for a review), Bertini et al. constructed the coarse-grained 'fluctuating hydrodynamic' limit of such diffusive systems. They showed how to construct a (WKB / Friedlin-Wentzel) theory where the amplitude of the effective noise is the small parameter. The problem maps then onto a Hamiltonian (or Hamilton-Jacobi) field theory in one space dimension, expressed in terms of a density field $\rho(x)$ and its conjugate $\hat{\rho}(x)$. The papers contain a second development that comes as a surprise: there is an explicit transformation that maps the evolution between a configuration $\rho_1(x, t')$ and $\rho_2(x, t)$ into one between $\rho_2(x, t')$ and $\rho_1(x, t)$ *even when the system is driven out of equilibrium by the baths*. Such transformations are available when there is a time-reversal symmetry - typically detailed balance - in systems with equilibrium baths. In order to explain this miracle, Tailleur et al. [6] showed that in fact, the large deviations around the hydrodynamic limit of SEP and of KMP could be transformed, via non-local *canonical* transformations, into equilibrium. The mapping from 'downhill' to 'uphill' is then nothing but the usual one, once the system is transformed into an undriven system with detailed balance. It was however never clear which systems enjoy such exceptional feature.

We shall show here that, for a large class of systems, there exists a mapping from the non-equilibrium process to the equilibrium process *already at microscopic level, and in any dimensionality*. The main result of this paper is then that diffusive systems driven in a non-equilibrium state by (multiple) external reservoirs are in direct relation with the corresponding equilibrium systems. See Figure 1 for a pictorial representation of the mapping between a system with three reservoirs at temperatures $T_1, T_2, T_3$ and a system with reservoirs all having the same temperature $T$.

Before going on, it is useful to mention that the particle transport models like SEP, and the energy transport models like KMP, may be treated in a unified group-theoretical way that uncovers their similarities – and indeed opens a whole set of connections with the work on quantum chains, especially those related to AdS/CFT and high-energy QCD [7–11].

It turns out that the mapping to equilibrium is not dependent on integrability, but it derives from a property extensively studied by probabilists: *duality* [12]. This property in turn is a consequence of a group of invariance of the generator of the evolution, that is by no means obvious when looking at the transition rules.

Although integrability is inessential, when the systems are integrable the mapping may be constructed explicitly using the conserved charges. As an example, we will work out in detail (cf. Section 4.3) the explicit formulas for the mapping for the SEP model following the recent results of [13].

The results of this paper may be extended to quantum stochastic systems driven in a non-equilibrium steady state by Lindblad reservoirs [14]. For the sake of clarity we restrict here to classical interacting particle systems and we shall discuss the quantum counterpart [15] in a forthcoming paper.

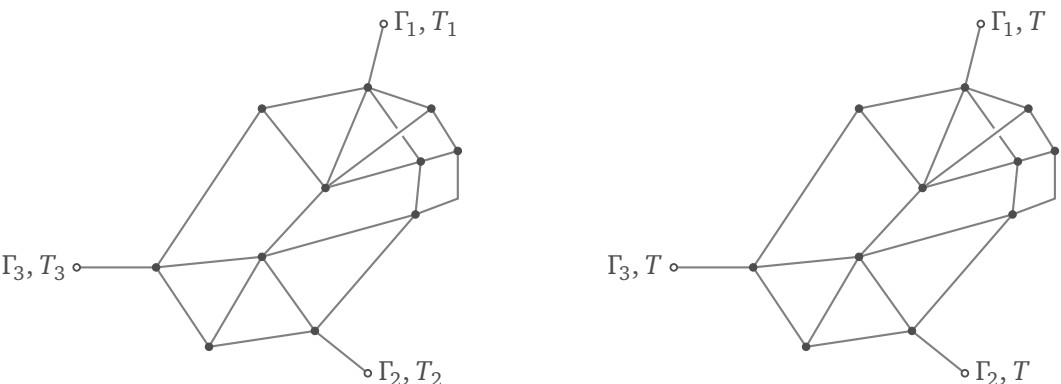

Figure 1: A probability distribution in the system of the left is mapped onto one in the system of the right. The mapping is preserved by subsequent evolution and may be inverted. The stationary state attained in the system of the left is mapped onto the equilibrium state of the system to the right. The couplings to the reservoirs $\Gamma_i$ remain the same.

## 2 Group-based diffusive models

Our results apply to a whole class of transport models, in particular the spatial structure does not matter. *The system is thus defined on a general graph $G = (V, E)$, not necessarily planar.* The graph has vertices $V = \{1, 2, \ldots, N\}$ and edges $E$. On the vertices (labelled by $i \in V$) we have, when the models are discrete, a given number of particles which may or may not be limited; otherwise, when the models are continuous, we have a continuous quantity ('energy'). In the edge set $E$, we distinguish:

- **Internal edges**, along which these quantities are transported, according to a certain probabilistic rule which may depend on the edge. We assume this rule is symmetric, i.e. the edges are unoriented.

- **External edges ('leads')**, connecting vertices to different *reservoirs*, which may be particle or thermal baths. The baths are defined by their thermodynamic properties (chemical potential $\mu_i$ or temperature $T_i$), and by their coupling strength $\Gamma_i \geq 0$. If the $\mu_i$ (or the $T_i$) are all the same, the system eventually reaches thermodynamic equilibrium. Otherwise, a stationary state with transport is eventually reached.

The most studied graphs of this kind are one-dimensional chains, but we shall not restrict to these here. The class of models that we can treat is identified by the following property: *they are diffusive models and they posses a symmetry group, yielding an algebraic description of the process generator.* On the surface, this looks quite restrictive and abstract, but the class actually contains several natural models considered so far in the mathematical study of transport. We defer the algebraic description, together with the identification of the symmetry group, to the next section. Here we introduce a few models representative of this class; we also recall their duals.

### 2.1 Particle transport models

**Symmetric partially excluded processes: SEP($n$).** These are systems of particles with repulsive interactions. In the most general version [16, 17] one assumes maximal occupancy $n \in \mathbb{N}$, which justifies the name SEP($n$). Particles evolve stochastically, following a sequence of jumps at (exponentially distributed) random times, as follows:

- Rule for transport from a vertex $k$ to a vertex $\ell$: the internal edge $(k, \ell)$ is chosen with rate $A_{k\ell} > 0$. The jumping rate to move one particle from $k$ to $\ell$ is proportional to the number of particles on the departure site $n_k$, and to the number of 'holes' $(n - n_\ell)$ in the arrival site $\ell$. Similarly for a jump from $\ell$ to $k$.

- Rule for particle reservoirs: a lead connected to site $i$ is chosen with rate $\Gamma_i > 0$. A particle is injected in $i$ with rate $\rho_i$ times the number of holes in $i$, and extracted from $i$ with rate $(1 - \rho_i)$ times the number of particles in $i$.

**Dual process.** For any maximal occupation number $n$ the model has a dual process [16, 18, 19] which is obtained by adding extra sites (one for each lead connected to reservoirs). The dual particles move on the internal edges like the particles in the original process did (self-duality). Furthermore, a dual particle sitting on a site that in the original process was connected to a lead, is absorbed at rate 1 in the corresponding extra site. As a consequence, the dual dynamics voids the sites of the graphs and all particles are eventually absorbed at the extra sites.

In this paper, to keep working always on the same Hilbert space, we will avoid the introduction of the extra sites. As we shall see, the price we will need to pay is a transformed Hamiltonian which is not stochastic, but only as an intermediate step.

## 2.2 Energy transport models

**KMP processes.** The first microscopic stochastic model of energy transport, in the context of Fourier's law, was introduced in 1982 by Kipnis, Marchioro and Presutti (KMP) [2], who studied a chain of oscillators randomly exchanging their energies between themselves and with two energy reservoirs at different temperatures. In the same spirit, we consider here a family of systems called (generalized) KMP processes [20]. The family is labelled by a positive real $s > 0$. In the course of time, the random jumps occur as follows:

- Rule for transport from a vertex $k$ to a vertex $\ell$: the internal edge $(k, \ell)$ is chosen with rate proportional to $A_{k\ell}$. Energies $z_k$ and $z_\ell$ of vertices $k$ and $\ell$ are randomly redistributed by allotting a fraction $B(z_k + z_\ell)$ on site $k$, and the remaining fraction $(1-B)(z_k + z_\ell)$ on site $l$. Here $B$ is a beta-distributed random variable with parameters $(2s, 2s)$, i.e. it is a law on in the interval $[0, 1]$ with probability density $f(b) \propto b^{2s}(1-b)^{2s}$. Thus the case $s = 1/2$ gives the uniform redistribution rule of the original KMP model.

- A lead $i$ is chosen with rate $\Gamma_i$. The energy of the site $i$ is refreshed with an equilibrium rule of temperature $T_i$ (heat bath or Monte Carlo).

**Dual process.** Similarly to the previous example, KMP processes have dual processes made of particles and absorbing extra sites [2, 19–21]. In the dual process, an internal edge $(k, \ell)$ is chosen with rate proportional to $A_{k\ell}$ and then, out of the available $n_k + n_\ell$ particles, a random number $M$ of them is put on site $k$ and the remaining number on site $\ell$. The random number $M$ has a beta-binomial distribution [1] with parameters $(n_k + n_\ell, 2s, 2s)$. The case $s = 1/2$ gives a uniform discrete redistribution of dual particles. Furthermore, a dual particle sitting on a site that in the original process was connected to a lead, is absorbed in the corresponding extra site at rate 1.

---

[1]The beta-binomial distribution with parameters $(n, a, b)$ gives the number of heads in $n$ Bernoulli experiments with a coin whose head probability is fixed but randomly drawn from a beta distribution with parameters $(a, b)$.

## 2.3   Other models

Other group-based diffusive models, that have physical properties similar to those discussed above, are introduced.

**Integrable models.**   Models that guarantee integrability play a special role as they offer the possibility of getting explicit formulas. In the framework of partially excluded processes of Section 2.1, the integrable model is obtained for $n = 1$, which gives the Symmetric Exclusion Process made of particles with hard core exclusion [12]. The non-equilibrium stationary state is constructed by the Matrix Product Ansatz [3], from which large deviations of the density and of the current may be computed, see [4] for a review of the exact solution. An alternative construction of the stationary measure was recently obtained in [13] using duality within the algebraic framework of the quantum inverse scattering method [22]. For the SEP model, the mapping from non-equilibrium to equilibrium has been described in [6] *at the hydrodynamic level*, by exploiting results from the macroscopic fluctuation theory [5]. As we will discuss in Section 5 the results of [13] allow for such mapping on the microscopic level.

As for the energy transport model of Section 2.2, none of the KMP processes is integrable. A family of *boundary-driven* integrable energy transport model has been recently introduced in [23] (see the discussion on Section 3.3 for the relation of these models to other models in the probability theory/field theory literature). For spin $s = 1/2$ the model is a Lévy process described as follows:

- On the internal edge $(k, \ell)$ with energies $z_k$ and $z_\ell$, an amount $\alpha$ of energy (with $0 < \alpha < z_k$) is moved from site $k$ to site $\ell$ as a Poisson process with intensity $\frac{d\alpha}{\alpha}$. Similarly for a jump from $\ell$ to $k$.

- At the sites $i$ connected to leads, jumps decreasing the energy by an amount $\alpha$ (with $0 < \alpha < z_i$) occur with intensity $\frac{d\alpha}{\alpha}$ and jumps increasing the energy by any amount $\alpha > 0$ occur with intensity $\frac{d\alpha}{\alpha} e^{-\lambda_i \alpha}$ with $\lambda_i > 0$.

The absorbing dual process of this model is made of particles with clustered jumps. Namely, if $n_i$ particles are sitting on site $i$ then a jump of $r$ particles (with $1 \leq r \leq n_i$) occurs at rate $1/r$. From a site $i$, the particles are moved, with the same probability, either to a neighbouring site (connected by an internal edge) or to the extra site (connected by lead). The general integrable model, labelled by spin value $s > 0$, is analyzed in [23].

**Diffusions.**   For energy transport models one can replace the evolution through Markovian jumps of the generalized KMP processes with a diffusions having continuous paths. This gives the so-called Brownian energy processes introduced in [20] or the Brownian momentum processes [21, 24]. The dual processes of those are called *inclusion processes* [25], they are the bosonic counterpart of partially excluded processes. More precisely, in the model with spin $s > 0$, the particles jump from vertex $k$ to a vertex $\ell$ with rate $n_k(2s + n_\ell)$.

**Inhomogeneiteis and/or multispecies.**   One can add inhomogeneities to allow different maximal occupancy at each site keeping the key duality property [26]. Several kinds of particles or continuous quantities, with mutual exclusion properties, are also possible, leading to higher rank algebras [27, 28].

# 3 Probability evolution generators

In this section we develop the algebraic description of the Markov processes discussed above. Such algebraic approach relies on applying to the equation describing the evolution of the probability for the system of interest – the so-called forward Kolmogorov equation – tools that are standard for the Schrödinger equation. It is well-known [16] that exclusion processes can be expressed in terms of a Hamiltonian (in fact, the generator of the probability evolution) with spin operators of the $\mathfrak{su}(2)$ Lie algebra. For the energy transport models it was realized in [20, 21] that also KMP processes can be described in this way, just replacing the compact spin algebra by the non-compact $\mathfrak{su}(1,1)$ algebra. See [29] for an extended presentation of the algebraic approach, and its use in finding dual processes.

## 3.1 Symmetric partially excluded processes

The evolution operator of these processes can be written as the Hamiltonian $H$ of the half-integer spin $j = n/2$ ferromagnet [16]

$$
\begin{aligned}
H &= \sum_{k,l \in V^2} A_{kl}\left(J_k^+ J_l^- + J_k^- J_l^+ + 2J_k^0 J_l^0 - 2j^2\right) \\
&+ \sum_{i \in V} \Gamma_i\left[\rho_i(J_i^+ + J_i^0 - j) + (1 - \rho_i)(J_i^- - J_i^0 - j)\right].
\end{aligned}
\tag{1}
$$

The operators $J_i^+, J_i^-, J_i^0$ act on the Hilbert space corresponding to $0 \le r \le n$ particles per site $i$ as follows:

$$
\begin{aligned}
J_i^+|r\rangle_i &= (2j - r)|r + 1\rangle_i \\
J_i^-|r\rangle_i &= r|r - 1\rangle_i \\
J_i^0|r\rangle_i &= (r - j)|r\rangle_i.
\end{aligned}
\tag{2}
$$

We order the states like $|2j\rangle, |2j-1\rangle, \ldots, |0\rangle$ so that the $J_i^-$ operators are lower-triangular. The Hamiltonian (1) acts on the tensor product space with states $\otimes_{i \in V}|r\rangle_i$ and it is easy to read off the (negative) transition rates. For instance the first term $-J_k^+ J_l^-$ gives the rate $n_\ell(2j - n_k)$ for the jump of a particle from site $\ell$ with $n_\ell$ particles to site $k$ with $n_k$ particles. The operators $J_i^+, J_i^-, J_i^0$ satisfy the commutation relations of the $\mathfrak{su}(2)$ Lie algebra:

$$
[J_i^0, J_i^\pm] = \pm J_i^\pm \qquad [J_i^-, J_i^+] = -2J_i^0.
\tag{3}
$$

Representations are labeled by the spin value $j$ related to the squared angular momentum operator via:

$$
\begin{aligned}
(\vec{J}_i)^2|j,m\rangle_i &= \left([J_i^0]^2 + \frac{1}{2}[J_i^+ J_i^- + J_i^- J_i^+]\right)|j,m\rangle_i \\
&= j(j + 1)|j,m\rangle_i.
\end{aligned}
\tag{4}
$$

Here $j = n/2$, so that the ordinary SEP with $(0,1)$ occupation corresponds to a representation of spin $j = 1/2$. For a given $j$ half-integer, there are $2j + 1$ eigenstates of the $J_i^0$ operator

$$
J_i^0|j,m\rangle_i = m|j,m\rangle_i,
\tag{5}
$$

with $m = r - j$ and $m \in \{-j, -(j-1), \ldots, j\}$. In other words, for a given half-integer $j$ we identify $|r\rangle_i = |j, r - j\rangle_i$, cf. (2).

## 3.2 KMP processes

We discuss here the diffusion processes associated to KMP, i.e. having the same algebraic structure. Moreover, following Kac's idea of energy transfer via random collisions [1], we think of kinetic energies, and thus discuss a velocity diffusion process where energy transfer is obtained by 'random rotations' of velocity vectors [30].

Thus we consider, on each site $i \in V$, a velocity vector with $M \in \mathbb{N}$ components, i.e. $v_{i,\alpha}$ with $\alpha = 1, \dots, M$. Suppose that they evolve as a diffusion process with Markov generator (acting on the core of $\mathscr{C}^\infty$ functions with compact support)

$$L = \sum_{k,\ell \in V^2} A_{k\ell} L_{k\ell} + \sum_{i \in V} \Gamma_i L_i, \tag{6}$$

where

$$L_{k\ell} = \sum_{\alpha,\beta=1}^{M} \left( v_{k,\alpha} \frac{\partial}{\partial v_{\ell,\beta}} - v_{\ell,\beta} \frac{\partial}{\partial v_{k,\alpha}} \right)^2, \qquad L_i = \sum_{\alpha=1}^{M} \left( T_i \frac{\partial^2}{\partial v_{i,\alpha}^2} - v_{i,\alpha} \frac{\partial}{\partial v_{i,\alpha}} \right).$$

Each term in $L_{k\ell}$ represents a rotation in the plane $(v_{k,\alpha}, v_{\ell,\beta})$, therefore it conserves the total kinetic energy of the edge $v_{k,\alpha}^2 + v_{\ell,\beta}^2$; on top of this, $L_i$ gives a Langevin bath with temperature $T_i$ and strength of coupling $\Gamma_i$.

The Fokker-Planck equation, yielding the evolution of the time-dependent probability density $p(v, t)$, reads

$$\frac{\partial p(v, t)}{\partial t} = L^* p(v, t), \tag{7}$$

where $L^*$ is the adjoint (in $\mathscr{L}^2(dv)$) of $L$. Expanding the brackets in (6) and defining

$$S_i^+ = \sum_{\alpha=1}^{M} \frac{1}{2} v_{i,\alpha}^2 \qquad S_i^- = \sum_{\alpha=1}^{M} \frac{1}{2} \frac{\partial^2}{\partial v_{i,\alpha}^2} \qquad S_i^0 = \sum_{\alpha=1}^{M} \frac{1}{4} \left( v_{i,\alpha} \frac{\partial}{\partial v_{i,\alpha}} + \frac{\partial}{\partial v_{i,\alpha}} v_{i,\alpha} \right)$$

we can rewrite this as an imaginary time Schrödinger equation with Hamiltonian $H = L^*$ and

$$L = \sum_{k,\ell \in V^2} A_{k\ell} \left( S_k^+ S_l^- + S_k^- S_l^+ - 2 S_k^0 S_l^0 + \frac{M^2}{8} \right) + \sum_{i \in V} \Gamma_i \left( T_i S_i^- - S_i^0 + \frac{M}{4} \right). \tag{8}$$

The generators $S^a$ satisfy the $\mathfrak{su}(1,1)$ Lie algebra relations:

$$[S_i^0, S_i^\pm] = \pm S_i^\pm \qquad [S_i^-, S_i^+] = 2 S_i^0. \tag{9}$$

Representations are labeled in an analogous manner by the spin value $s > 0$ with the Casimir operator:

$$
\begin{aligned}
(\vec{S}_i)^2 |s, m\rangle_i &= \left( [S_i^0]^2 - \frac{1}{2} [S_i^+ S_i^- + S_i^- S_i^+] \right) |s, m\rangle_i \\
&= s(s-1) |s, m\rangle_i.
\end{aligned}
\tag{10}
$$

Note the sign difference with respect to $\mathfrak{su}(2)$. To identify the representation (i.e. the value of $s$), we compute the eigenvalue of $S^0$ and $(\vec{S})_i^2$ when acting on the (constant) lowest weight state $|-\rangle_i$, which is the eigenvector with zero eigenvalue of $L^*$:

$$S_i^0 |-\rangle_i = \frac{M}{4} |-\rangle_i, \tag{11}$$

$$(\vec{S}_i)^2 |-\rangle_i = \frac{M}{4} \left( \frac{M}{4} - 1 \right) |-\rangle_i. \tag{12}$$

Hence $s = M/4$, and in particular $s = 1/4$ for the process with one velocity per site.

### 3.3 Integrable version of the energy transport model

For integrable transport models one may perform explicitly the mapping from non-equilibrium to equilibrium. However, in principle integrability is not necessary for the mapping to exist.

Once one identifies the KMP energy diffusion model with an $\mathfrak{su}(1,1)$ chain, the question immediately arises as to its integrability. While the chain with Hamiltonian (8) is not integrable, its integrable cousin has been known for a long time [31]. It remains to check that such a model may indeed be interpreted as a stochastic system, which it is indeed [23]. It turns out that the *particle version* of such a system has also been introduced in the probability literature, in the asymmetric form, by Sasamoto-Wadati [32] for spin $s = 1/2$ and by Povolotsky [33] and Barraquand-Corwin [34] for higher spin, just constructing it on the basis of the Bethe ansatz properties – without making a connection with integrable spin chain explicit. For the latter we refer the reader to [23] and [35].

The boundary driven integrable version of the energy transport model recently introduced in [23] is the Lévy process described in Section 2.3. In the corresponding particle version, the probability evolves with a generator of the form (6) where now

$$L_{k,\ell} = 2(\psi(\mathbb{S}_{k,\ell}) - \psi(2s)), \tag{13}$$
$$L_i = e^{\beta_i S_i^+} e^{\frac{1}{\beta_i - 1} S_i^-} \psi(S_i^0 + s) e^{-\frac{1}{\beta_i - 1} S_i^-} e^{-\beta_i S_i^+} - \psi(2s).$$

Here $\psi$ is the digamma function (the logarithmic derivative of the gamma function), $\beta_i$ is a parameter tuning the density of the reservoirs and the operator $\mathbb{S}_{k,\ell}$ is related to the two-site Casimir acting on two sites $k$ and $\ell$ via $(\vec{S}_k + \vec{S}_\ell)^2 = \mathbb{S}_{k,\ell}(\mathbb{S}_{k,\ell} - 1)$.

## 4 Duality transformations

As discussed in previous sections, dual Markov processes of transport models are usually obtained by introducing extra sites, one for each lead with $\Gamma_i > 0$. The key property of the dual process is the absorbing property of the extra sites, which is reflected in the fact that the dual generator has a *triangular structure*. Here we fully exploit the consequences of this crucial property. To keep working always on the same Hilbert space we avoid the extra sites, yielding a non-stochastic transformed Hamiltonian.

### 4.1 Microscopic mapping

Let us for definiteness concentrate on the SEP($n$) case (the same reasoning can be repeated in the other cases). As pointed out by Schütz and Sandow [16], one can make a "duality" transformation of the Hamiltonian $H$ in (1) as follows. From the Hadamard formula for the $\mathfrak{su}(2)$ algebra one obtains

$$\begin{aligned} e^{\mu J^\pm} J^0 e^{-\mu J^\pm} &= J^0 \mp \mu J^\pm \\ e^{\mu J^\pm} J^\mp e^{-\mu J^\pm} &= J^\mp \pm 2\mu J^0 - \mu^2 J^\pm. \end{aligned} \tag{14}$$

Setting $\mu = 1$, we get that the boundary terms of (1) transform into

$$e^{J_i^+} \Gamma_i \left[ \rho_i(J_i^+ + J_i^0 - j) + (1-\rho_i)(J_i^- - J_i^0 - j) \right] e^{-J_i^+} = \Gamma_i \left[ (J_i^0 - j) + (1-\rho_i)J_i^- \right]. \tag{15}$$

This is essentially equivalent to the general notion of duality of Markov processes [20, 29]. Indeed (15) can be rewritten as

$$\left[ \rho_i(J_i^+ + J_i^0 - j) + (1-\rho_i)(J_i^- - J_i^0 - j) \right]^{tr} R_i e^{-J_i^+} = R_i e^{-J_i^+} \left[ \rho_i J_i^- - J_i^0 - j \right], \tag{16}$$

where $^{tr}$ denotes transposition and $R_i$ is the diagonal matrix (with entries the inverse of the Binomial distribution) such that $R_i J_i^\pm R_i^{-1} = J_i^\mp$. If we introduce an extra site, called $c(i)$ and associated to site $i$, this can be further rewritten as

$$H_i^{tr} D_i = D_i H_i^{dual}, \tag{17}$$

where the duality function $D_i = \sum_m \rho_i^m \langle m | R_i e^{J_i^+}$, and $H_i^{dual} = a_{c(i)}^+ J_i^- - J_i^0 - j$ is the stochastic Hamiltonian that describes the dual absorbing process at site $i$ (here $a_{c(i)}^+$ is a bosonic creation operator in the extra site). Equation (17) is the standard form of a duality relation between two Markov processes [12, 29], that allows to connect expectations of the original process to expectation of the dual process. For a general discussion on the relation between duality relation and symmetries we refer the reader to [20] and reference therein; for a constructive approach aiming to introducing group-based models with duality we refer to [36].

Transforming further, with an arbitrary $0 < \bar{\rho} < 1$, we have

$$e^{-(1-\bar{\rho})J_i^-} e^{J_i^+} \Gamma_i \left[ \rho_i(J_i^+ + J_i^0 - j) + (1 - \rho_i)(J_i^- - J_i^0 - j) \right] e^{-J_i^+} e^{(1-\bar{\rho})J_i^-} =$$
$$= \Gamma_i \left[ (J_i^0 - j) + (\bar{\rho} - \rho_i)J_i^- \right]. \tag{18}$$

All in all, defining $J_{tot}^\pm = \sum_{i \in V} J_i^\pm$ we get that the original Hamiltonian (1) is mapped into:

$$H_B = e^{-(1-\bar{\rho})J_{tot}^-} e^{J_{tot}^+} H e^{-J_{tot}^+} e^{(1-\bar{\rho})J_{tot}^-} = H_0 + \underbrace{\sum_{i \in V} \Gamma_i(\bar{\rho} - \rho_i)J_i^-}_{B}. \tag{19}$$

This defines $B$ as the underbraced term and $H_0$ as

$$H_0 = \sum_{k,l \in V^2} A_{kl} \left( J_k^+ J_l^- + J_k^- J_l^+ + 2J_k^0 J_l^0 - 2j^2 \right) + \sum_{i \in V} \Gamma_i(J_i^0 - j). \tag{20}$$

In $H_0$, the bulk term of (1) is left invariant, precisely because of the global $\mathfrak{su}(2)$ symmetry. If all the $\rho_i = \bar{\rho}$ are the same, we may eliminate the $B$ term. *Note that $H_B$ does not correspond to a probability conserving process.*

Since the Hamiltonian $H_0$ commutes with $J_{tot}^0$, we may simultaneously diagonalize the two operators so that $J_{tot}^0 |\Lambda_k, m_{tot}\rangle = m_{tot}|\Lambda_k, m_{tot}\rangle$ and $H_0 |\Lambda_k, m_{tot}\rangle = \Lambda_k |\Lambda_k, m_{tot}\rangle$ where $k = 1, \ldots, (n+1)^{|V|}$ and $m_{tot}$ labels the corresponding number of particles. If we order this set of vectors by decreasing values of $m_{tot}$, and since each $J_i^-$ lowers the value of $m_{tot}$ by one, the matrix elements of $B$ are lower diagonal in this base. Hence it becomes apparent that the $J_i^-$ in $B$ do not modify the spectrum. We conclude that the spectra of $H_B$ and $H$ coincide with that of $H_0$. We remark that such isospectrality property for spin chain Hamiltonians differing by operators that are lower triangular in the right basis was also observed in [37]. By inspection of (20) we see that the spectrum depends only on the $\Gamma_i$'s, but not on the $\rho_i$'s. We conclude that there exists an operator $W$ such that

$$H_B = W H_0 W^{-1}. \tag{21}$$

In particular, denoting $|R_0^k\rangle \equiv |\Lambda_k, m_{tot}\rangle$, $\langle L_0^k|$ and $|R_B^k\rangle$, $\langle L_B^k|$ the right and left eigenvectors of $H_0$ and $H_B$, and $\Lambda_k$ the corresponding eigenvalues labelled by $k$, we have the relation

$$W |R_0^k\rangle = |R_B^k\rangle. \tag{22}$$

The transformation $W$ can be written as

$$W = \lim_{\epsilon \to 0} \epsilon \sum_k [\Lambda_k - H_B + \epsilon]^{-1} |R_0^k\rangle \langle L_0^k|. \tag{23}$$

Alternatively, one may also write an equivalent expression that, introducing an additional integral, does not require the knowledge of the spectrum of $H_B$:

$$W = \lim_{\epsilon \to 0} \epsilon \lim_{\bar{\epsilon} \to 0} \mathfrak{I} \oint d\Lambda \left[\Lambda - H_B + \epsilon\right]^{-1} \left[\Lambda - H_0 + i\bar{\epsilon}\right]^{-1}. \tag{24}$$

Here $\mathfrak{I}$ denotes the imaginary part. This expression in fact is essentially formal and it does not seem to be useful at this stage.

To recap, we have up to now a similarity transformation relating $H$ to $H_0$:

$$W^{-1} e^{-(1-\bar{\rho})J_{tot}^-} e^{J_{tot}^+} H e^{-J_{tot}^+} e^{(1-\bar{\rho})J_{tot}^-} W = H_0. \tag{25}$$

One further transformation gives the final result

$$P \, H \, P^{-1} = H_{eq}, \tag{26}$$

with

$$P = e^{-J_{tot}^+} e^{(1-\bar{\rho})J_{tot}^-} W^{-1} e^{-(1-\bar{\rho})J_{tot}^-} e^{J_{tot}^+} \tag{27}$$

and $H_{eq}$ denoting the Hamiltonian (1) in equilibrium with $\rho_i = \bar{\rho}$ for all $i \in V$. Thus we found a mapping of the non-equilibrium process $H$ (with chemical potentials $\rho_i$) to the equilibrium process with Hamiltonian $H_{eq}$ with reservoirs having the same chemical potential $\bar{\rho}$ at all leads! The coupling intensities $\Gamma_i$ of $H$ and $H_{eq}$ remain the same. If $|\psi\rangle$ is the ket vector which encodes the probability distribution of the original process as $\frac{d}{dt}|\psi\rangle = H|\psi\rangle$, and $|\psi\rangle_{eq}$ is the ket vector which encodes the probability distribution of the transformed process as $\frac{d}{dt}|\psi\rangle_{eq} = H_{eq}|\psi\rangle_{eq}$ then, as consequence of (26), one has $|\psi\rangle = P^{-1}|\psi\rangle_{eq}$. It would be interesting to study the relation of our approach to the matrix product ansatz [3] and [44–46].

For later use, it is convenient to define $A$ by

$$W = e^A. \tag{28}$$

The transformation (26) may be written in terms of $A$ as

$$e^{-[J_{tot}^+,\ ]} e^{(1-\bar{\rho})[J_{tot}^-,\ ]} e^{-[A,\ ]} e^{-(1-\bar{\rho})[J_{tot}^-,\ ]} e^{[J_{tot}^+,\ ]} H = H_{eq}, \tag{29}$$

where we used the notation $e^{[X,\ ]}Y = e^X Y e^{-X}$.

## 4.2 Perturbative approach

For all systems, the expression for $W$ we introduced in the previous paragraph may be evaluated perturbatively. To this aim we insert $\Delta$, a bookkeeping parameter we shall set to one at the end, to order the perturbation series. We start by writing

$$H_\Delta = H_0 + \Delta B, \tag{30}$$

and define $A_\Delta$ by

$$W_\Delta = e^{A_\Delta}, \tag{31}$$

which reduce to $H_B$ and $W$ for $\Delta = 1$ respectively. The relation between $H_B$ and $H_0$ in (21) becomes

$$e^{-A_\Delta} \left[H_0 + \Delta B\right] e^{A_\Delta} = e^{-[A_\Delta,\ ]} H_\Delta = H_0. \tag{32}$$

Then we may write a power series for $A_\Delta$ by putting $A_\Delta = \Delta A^{(1)} + \Delta^2 A^{(2)} + ...$, so that

$$
\begin{aligned}
B &= [A^{(1)}, H_0] \\
\frac{1}{2}[B, A^{(1)}] &= [A^{(2)}, H_0] \\
&\vdots
\end{aligned}
\tag{33}
$$

or, in the basis where $H_0$ is diagonal

$$
\begin{aligned}
B_{ij}(\Lambda_i - \Lambda_j)^{-1} &= A_{ij}^{(1)} \\
\frac{1}{2}[B, A^{(1)}]_{ij}(\Lambda_i - \Lambda_j)^{-1} &= A_{ij}^{(2)} \\
&\vdots
\end{aligned}
\tag{34}
$$

This allows to compute the coefficients $A^{(i)}$ of $W_\Delta$ recursively.

## 4.3  Integrable models

As we have seen in the previous sections, of all the transformations we need to perform, the only one that is non-local in the sites, and does not have an explicit expression, is $W$. Consider now the expressions for the matrix $W$ (23): it depends on $H_0$ and $H_B$. It is easy to see that if we have an operator $Q_B$ (conserved charge) that commutes with $H_B$, and an operator $Q_0$ that commutes with $H_0$, with $Q_0$ and $Q_B$ iso-spectral, we may just as well substitute the $H$'s by the $Q$'s in (23). The change of basis induced by $W$ will be the same. If, furthermore, the spectrum of $Q_0$ is known explicitly, we may obtain an important simplification and sum up the perturbative series. *This is what integrability gives us.*

We illustrate this procedure in the prototype case of boundary-driven SEP on a 1D chain of $N$ sites. In the bulk, particles jumps at rate 1 to their nearest neighbors, provided there is space in the arrival site. The reservoir connected to the first site (resp. last site) inject particles at rate $\alpha$ (resp. $\delta$) and remove particles at rate $\gamma$ (resp. $\beta$). We consider the Hamiltonian (1) with $j = 1/2$ and define $J_i^\pm = \sigma_i^\pm$, $J_i^0 = \sigma^0/2$ with $\sigma_i^\pm = (\sigma_i^x \pm \sigma_i^y)/2$ and $\sigma_i^0 = \sigma_i^z$ where $\sigma_i$ are the Pauli matrices at site $i$, so that

$$
\begin{aligned}
H &= \sum_{i=1}^{N-1}\left(\sigma_i^+\sigma_{i+1}^- + \sigma_i^-\sigma_{i+1}^+ + \frac{1}{2}\sigma_i^0\sigma_{i+1}^0 - \frac{1}{2}\right) \\
&+ \sum_{i\in\{1,N\}} \Gamma_i\left[\rho_i(\sigma_i^+ + \sigma_i^0/2 - 1/2) + (1-\rho_i)(\sigma_i^- - \sigma_i^0/2 - 1/2)\right],
\end{aligned}
\tag{35}
$$

with $\Gamma_1 = \alpha + \gamma$, $\Gamma_N = \delta + \beta$, $\rho_1 = \alpha/(\alpha + \gamma)$, $\rho_N = \delta/(\delta + \beta)$. It is convenient to set $\bar{\rho} = \rho_N$ so that, after performing the transformation in (19), one gets $H_B = H_0 + B$ with

$$
H_0 = \sum_{i=1}^{N-1}\left(\sigma_i^+\sigma_{i+1}^- + \sigma_i^-\sigma_{i+1}^+ + \frac{1}{2}\sigma_i^0\sigma_{i+1}^0 - \frac{1}{2}\right) + \sum_{i\in\{1,N\}}\frac{\Gamma_i}{2}\left(\sigma_i^0 - 1\right)
\tag{36}
$$

and

$$
B = \Gamma_1(\rho_N - \rho_1)\sigma_1^-.
\tag{37}
$$

The study of the Hamiltonian $H_0$ using the coordinate Bethe ansatz goes back to [38,39] and to Sklyanin [22] using the quantum inverse scattering method. For its relation to the Hamiltonian of the SEP within the latter framework we refer the reader to [40–42]. The isospectrality of the Hamiltonians (35) and (36) is straight-forwardly obtained. The transformation $W$, cf. Section 4.1, however was only obtained recently [13]. Without going into the details of the quantum inverse scattering method [31], which allows to obtain the conserved charge, we take the operator $Q_B$ as given and show how the results of [13] fit into our perturbative framework. We further present some alternative ways of writing the resulting similarity transformation, see in particular (47) and (51).

As mentioned before, the explicit computation of the similarity transformation $W$ relies on the existence of a "charge" operator

$$
Q_\Delta = Q_0 + \Delta Q_-,
\tag{38}
$$

that commutes with the Hamiltonian $H_\Delta$ for any $\Delta$, i.e.

$$[H_\Delta, Q_\Delta] = 0. \tag{39}$$

The operator $Q_\Delta$ has the same eigenvectors as the Hamiltonian $H_\Delta$ which reduce to the eigenvectors of $H_0$ (or $Q_0$) for $\Delta = 0$ and to the ones of $H_B$ for $\Delta = 1$. Thus the transformation $W_\Delta$ maps $Q_0$ to $Q_\Delta$:

$$Q_\Delta = W_\Delta Q_0 W_\Delta^{-1}, \tag{40}$$

cf. (32). The operator $Q_\Delta$ follows from the transfer matrix of the integrable spin chain, see [13] for further details. The unperturbed part only depends on the total spin

$$Q_0 = \frac{\sigma_{tot}^0}{2} \Gamma_1 \Gamma_N \left( \frac{\sigma_{tot}^0}{2} + \Gamma_1^{-1} + \Gamma_N^{-1} \right), \tag{41}$$

while the lower triangular part is given by the non-local expression

$$Q_- = \Gamma_1(\rho_N - \rho_1) \left( \sigma_{tot}^- + \Gamma_N \sum_{i=1}^{N} \sigma_i^- \left( \frac{\sigma_i^0 - 1}{2} + \sum_{k=i+1}^{N} \sigma_k^0 \right) \right). \tag{42}$$

The similarity transformation can then be computed using the algebraically more simple operator $Q_\Delta$:

$$W_\Delta = \lim_{\epsilon \to 0} \epsilon \sum_m \left[ \Lambda_m^Q - Q_\Delta + \epsilon \right]^{-1} |R_0^m\rangle \langle L_0^m|. \tag{43}$$

The eigenvalues $\Lambda_m^Q$ can be read off immediately from the explicit form of $Q_0$ in (41). One way to determine the $W_\Delta$ is by recursively solving (40) in powers of $\Delta$ as discussed in Section 4.2. Writing

$$W_\Delta = 1 + \sum_{k=1}^{N} \Delta^k g_k \tag{44}$$

we get

$$g_k Q_0 = Q_0 g_k + Q_- g_{k-1} \qquad k = 1, \ldots N, \tag{45}$$

with $g_0 = 1$. From this it follows that $g_k$ must annihilate $k$ particles which implies its exchange relation with $Q_0$:

$$g_k Q_0(\sigma_{tot}^0) = Q_0(\sigma_{tot}^0 + 2k) g_k.$$

This allows us to solve the recursion and explicitly write $g_k$ in terms of $Q_0$ and $Q_-$. By using then the definition of $Q_0$ (41) we find

$$W_\Delta = \sum_{k=0}^{N} \frac{\Delta^k}{k!} \left( \frac{Q_-}{\Gamma_1 \Gamma_N} \right)^k \frac{\Gamma\left(\sigma_{tot}^0 + \Gamma_1^{-1} + \Gamma_N^{-1} - k\right)}{\Gamma\left(\sigma_{tot}^0 + \Gamma_1^{-1} + \Gamma_N^{-1}\right)}. \tag{46}$$

The final expression for $W$ can be obtained by taking $\Delta = 1$. From this expression, the non-local character of the $W$-transform is clearly seen, cf. (42).

Curiously, $W$ can be written as an exponential function

$$W = \exp\left( \sum_{k=1}^{N} A^{(k)} \right), \tag{47}$$

with

$$A^{(k)} = \gamma_k \frac{(-1)^{k+1}}{k! \, \Gamma_1^k \Gamma_N^k} \frac{\Gamma\left(1 + \sigma_{tot}^0 + \Gamma_1^{-1} + \Gamma_N^{-1}\right)}{\Gamma\left(2k + \sigma_{tot}^0 + \Gamma_1^{-1} + \Gamma_N^{-1}\right)} Q_-^k, \tag{48}$$

where the function $\gamma_k$ are related to the Clebsch-Gordan coefficients (compare [43]), $\gamma_k = (1, 1, 3, 14, 80, 468, 2268, 10224, 313632, 9849600, \ldots)$. This exponential form of $W$ allows to read off its inverse.

The expression for $W$ may be expressed in terms of Bessel function which may be more suitable to take the hydrodynamic limit. To see this, we define $P_j$ the projector on the subspace of total spin $j$. Then we may rewrite (46) for $\Delta = 1$ as

$$W = \sum_j \sum_{k=0}^{N} \frac{1}{k!} \left( \frac{Q_-}{\Gamma_1 \Gamma_N} \right)^k \frac{\Gamma\left(2j + \Gamma_1^{-1} + \Gamma_N^{-1} - k\right)}{\Gamma\left(2j + \Gamma_1^{-1} + \Gamma_N^{-1}\right)} \, P_j \,. \tag{49}$$

Now, writing the gamma function in the numerator as $\Gamma(z) = \int dx \, x^{z-1} e^{-x}$ we may perform the sum over $k$, to obtain:

$$W = \sum_j \int dx \, e^{-x + \frac{Q_-}{\Gamma_1 \Gamma_N} x^{-1}} \, x^{\left(2j + \Gamma_1^{-1} + \Gamma_N^{-1} - 1\right)} \frac{1}{\Gamma\left(2j + \Gamma_1^{-1} + \Gamma_N^{-1}\right)} \, P_j \,. \tag{50}$$

We recognize the generator of the Bessel function $K_n(z)$, so that we may write:

$$W = 2 \sum_j \frac{K_{-2j - \Gamma_1^{-1} - \Gamma_N^{-1}} \left[ 2\sqrt{\frac{-Q_-}{\Gamma_1 \Gamma_N}} \right]}{\Gamma\left(2j + \Gamma_1^{-1} + \Gamma_N^{-1}\right)} \left( \sqrt{\frac{-Q_-}{\Gamma_1 \Gamma_N}} \right)^{2j + \Gamma_1^{-1} + \Gamma_N^{-1}} P_j \,. \tag{51}$$

This is an explicit expression for $W$ acting on a function of given $\sigma_{tot}^0$.

One final remark: when we map a probability distribution via this transformation, we will not necessarily obtain a function that is positive definite, and may be interpreted as a probability. This is no real problem if the system is ergodic: one may add to the function an equilibrium distribution multiplied by a sufficiently large factor, and normalize the result to one. The evolution of this linear combination, which may be interpreted as a bona-fide probability, is just a linear combination of the one of the original vector and the equilibrium one, and the latter does not evolve.

## 5 Fluctuating hydrodynamics

Fluctuating hydrodynamics is obtained by coarse graining a system: the quantities obtained depend on space and have fluctuations that are the smaller, the larger the coarse graining. The hydrodynamic limit consists formally of two parts: one first goes to a continuous chain, and then performs the long-wavelength (infrared) limit. These cases amount to a 'semiclassical' approximation in the following sense: just as the probability evolution can be seen as a Schrödinger equation with imaginary time, its coarse-grained limit correspond to the semiclassical limit of it – the role of $\hbar$ being played by the intensity of fluctuations. The (WKB) treatment of this with the usual classical tools is variously known as 'Freidlin-Wentzel' or 'Hamilton-Jacobi' approach in the mathematical physics literature. Exactly the same problem arises in the AdS/CFT literature, where the long-wavelength limit becomes the high angular momentum regime of the string [47]. Let us stress again the role of group theoretical nature of these problems: it is because of this structure that 'coarse graining' amounts to 'large spin representation', and through it the semiclassical limit.

At the fluctuating hydrodynamics level, the transformations connecting non-equilibrium to equilibrium can be read from the previous Section 4.1; just changing commutators into Poisson brackets:

$$\{\mathcal{J}_k^0, \mathcal{J}_k^\pm\} = \pm \mathcal{J}_k^\pm \qquad \{\mathcal{J}_k^-, \mathcal{J}_k^+\} = -2\mathcal{J}_k^0 \,. \tag{52}$$

For the one-dimensional SEP(2j), a realisation of (52) can be obtained starting from a representation of the $\mathfrak{su}(2)$ algebra on the lattice, cf. [6]:

$$J_k^+ = (2j - \rho_k)e^{\hat{\rho}_k}, \qquad J_k^- = \rho_k e^{-\hat{\rho}_k}, \qquad J_k^0 = -(\rho_k + j), \tag{53}$$

where $\hat{\rho}_k = \frac{\partial}{\partial \rho_k}$. Redefining $\rho_k \to \frac{\rho_k}{2j}$, so that the commutator is $\frac{1}{2j}$, and going to the continuous limit, in terms of a density variable $\rho(x)$ and its conjugate field $\hat{\rho}(x)$, then (53) becomes:

$$\mathcal{J}^+ = 2j(1 - \rho(x))e^{\hat{\rho}(x)}, \qquad \mathcal{J}^- = 2j\rho(x)e^{-\hat{\rho}(x)}, \qquad \mathcal{J}^0 = -2j(\rho(x) + 1/2). \tag{54}$$

We obtain the small-noise, hydrodynamic limit as a 'semiclassical' one in the usual way. The transition probability between $\rho(x, t_1)$ and $\rho(x, t_2)$ reads (see [5, 18, 48])

$$P(\rho(x, t), \rho(x, t')) = \int \mathcal{D}\hat{\rho}\mathcal{D}\rho \, e^{-2jNS[\rho, \hat{\rho}]}, \tag{55}$$

$$S[\rho, \hat{\rho}] = \int_{t_1}^{t_2} \int_0^1 dx dt \{\hat{\rho}\partial_t \rho - \mathcal{H}[\rho, \hat{\rho}]\}. \tag{56}$$

This is a path integral with boundaries $\rho(x, t_1)$ and $\rho(x, t_2)$ with 'classical' Hamiltonian

$$\mathcal{H}[\rho, \hat{\rho}] = \frac{1}{2}\sigma(\rho)(\nabla\hat{\rho})^2 - \frac{1}{2}\nabla\rho\nabla\hat{\rho}, \tag{57}$$

where $\sigma(\rho) = \rho(1 - \rho)$. The fields $\rho$ are constrained, in the hydrodynamic limit, to satisfy the spatio-temporal boundary conditions

$$\forall t, \qquad \rho(0, t) = \rho_0, \quad \rho(1, t) = \rho_1. \tag{58}$$

The quantum model suggests that, at the fluctuating hydrodynamics level, the mapping (29) between non-equilibrium and equilibrium is replaced by

$$e^{-\{\mathcal{J}_{tot}^+,\}}e^{(1-\bar{\rho})\{\mathcal{J}_{tot}^-,\}}e^{-\{\mathcal{A},\}}e^{-(1-\bar{\rho})\{\mathcal{J}_{tot}^-,\}}e^{\{\mathcal{J}_{tot}^+,\}}\mathcal{H} = \mathcal{H}_{eq}, \tag{59}$$

where $\mathcal{H}_{eq}$ coincides in form with $\mathcal{H}$ but the field $\rho'$ satisfies the boundary conditions

$$\forall t, \qquad \rho'(0, t) = \rho'(1, t) = \bar{\rho}. \tag{60}$$

This is a *canonical* transformation that may be seen as mapping $(\rho(x), \hat{\rho}(x))$ into another pair $(\rho'(x), \hat{\rho}'(x))$, but not a simple *contact* transformation mapping $\rho(x)$ into another pair $\rho'(x)$.

Now, in principle $\mathcal{A}$ can be obtained with the classical perturbation equations:

$$\mathcal{B} + \{\mathcal{H}_0, \mathcal{A}^{(1)}\} = 0$$
$$\frac{1}{2}\{\mathcal{B}, \mathcal{A}^{(1)}\} + \{\mathcal{H}_0, \mathcal{A}^{(2)}\} = 0 \tag{61}$$
$$\vdots$$

Here $\mathcal{A}, \mathcal{H}_0$ and $\mathcal{B}$ denote the 'semiclassical' counterparts of $A, H_0$ and $B$ introduced in the microscopic model.

One has to solve these equations under the assumption that $\mathcal{A}^{(r)}$ are functions of the spins, or of $\rho(x), \hat{\rho}(x)$. This will have a solution to the extent that $\mathcal{H}_0$ and $\mathcal{H}_\mathcal{B}$ are *classically integrable*. In that case, one may take advantage of the fact that $\mathcal{H}_0$ may be written in action-angle variables. It turns out that the fluctuating hydrodynamic limit equations for all the models considered here are classical integrable models. One may understand this by arguing that several models (e.g. all the partial exclusion models) share a common hydrodynamic limit with an integrable one (in this case the SEP). The same is true, for example, with the KMP model, which is not integrable, but shares the same hydrodynamic limit as the model of (13). In models that are not classically integrable, the classical perturbation would be applied on a trajectory – the instanton in this case.

## 6 Discussion

The results in this paper have to be seen in the light of the work of Graham [49], who long time ago observed, in the case of Langevin dynamics, that the distinction between processes that do satisfy reversibility and processes that do not is somewhat artificial. Our results are in this vein, although here the mapping is of the system driven out of equilibrium to *the same* system with equilibrium baths.

As mentioned above, the transformation mapping the driven problem into the undriven one is *not* a function between particle occupations of the two respective systems, but it involves the whole set including the auxiliary 'spin' variables. For example, in a problem where occupation numbers are associated with the spins $s_i^0$, the transformation is not only non-local in the site $i$ of the form $\{s_i^0\} \rightarrow \{s_j'^0\}$, but it involves all the spin operators on a site: in other words, *the mapped occupation numbers are a function, not only of the original occupation numbers, but also of the current operators on each link*. This explains an apparent paradox about Onsager's principle: in the driven problem closed trajectories form a loop, while in the undriven one they do not. A pure 'contact' transformation can only transform loops into loops.

The mapping $P$ in (26) allows one to obtain the non-equilibrium steady state by transforming the equilibrium measure. At the hydrodynamic level, the mapping reproduces the non-equilibrium free energy found as the density large deviation function in MFT [5, 6, 50].

A recurrent question in driven out of equilibrium problems concerns the existence of an 'out of equilibrium free energy'. Here we have, for the driven system, a distinguished candidate: it is the (usual) free energy of the mapped system. This may in principle be written as a function of the old density fields *and, from the discussion above, their time derivatives*. All in all, we believe that this angle of study of these problems provides an excellent framework in which one may discuss what properties one can expect, and which ones one cannot expect, from problems driven out of equilibrium.

## Acknowledgments

We thank the anonymous referees for their comments.

**Funding information** J.K. is supported by the Simons Foundation Grant No 454943. R.F. is supported by the German research foundation (Deutsche Forschungsgemeinschaft – DFG) Research Fellowships Programme 41652715.

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
