# Peer review of "Duality and hidden equilibrium in transport models"

_SciPost Physics, doi:SciPost Phys. 9, 054 (2020)_

## Round 2 · Referee Report · Anonymous · 2020-7-17

Report

Dear Editor, dear Authors

I read with interest the manuscript Duality and hidden equilibrium in transport models. The Authors discuss several examples of transport models that admit a similarity transformation of the generator into an operator satisfying detailed balance, arguing that this transformation is possible whenever a general notion of “duality” is present - rather than the more restrictive integrability usually considered in the specialized literature.

It is a nicely written remarkable paper. The ideas presented in the introductory parts of the paper are very broad and intriguing, and the deployment of technical ability impressive. However, the development is challenging to the non-expert, and it is not clear to what degree the ideas proposed in the introduction are proven and motivated in full generality, given that the paper mostly focuses on specific models.

More precisely:

A general notion of duality is mentioned but never introduced in a general setting, apart from stating that it is a well-known duality in probability theory. Also, it is not clear how the existence of a symmetry group for the generator interplays with such duality. Is there a general way to introduce group-based diffusive models and duality independently of the specific cases analyzed?

The technical knowledge displayed of the rich mathematics of interacting particle models is impressive, but at times it looks like a piece of virtuosism (for example: what is the actual need for eq. 4.9? Also: in Sec: 4.3 it is mentioned that, quite intriguingly, integrability allows to solve for W whenever Q_0 is easier to diagonalize than H_0: but it appears that in the following treatment an expression for W is found without diagonalizing H_0)

In the introduction it is mentioned that there exist a time-reversed mapping between \rho_2 and \rho_1 even when the system is driven out of equilibrium. But how is this special of equilibrium systems? Given a MC propagator e^tH one can always define the inverse map e^-tH. Detailed balance has more to do with correlation functions than with probabilities.

It is mentioned in passing that the new detailed balanced dynamics has absorbing sites. But then, the stationary state of the new equilibrium dynamics will be different from the stationary state of the original dynamics. So, while there is a mapping between generators, it appears that there is no mapping between the time-evolved probabilities, and that stationary states map to specific transient states. Maybe the Authors should explain better what exactly the mapping consists of.

Best regards

The referee

---

## Round 2 · Referee Report · Anonymous · 2020-7-28

Report

This is a very interesting paper, a long awaited (for me at least) follow up to work by some of the authors over a decade ago on mapping between driven stochastic systems and equilibrium ones. The paper thus addresses a very important question in statistical mechanics, that of the fundamental distinction between non-equilibrium and equilibrium.

The setting is that of transport processes connected to leads, like the SEP and its generalisations. Bulk conservation allows to describe these models in hydrodynamic terms in the appropriate scaling limit via MFT. This is the usual framework to study non-equilibrium transport and associated questions. In such settings what is being transported is conserved (let's say carries energy) so individual transitions in the bulk are reversible (have equal probabilities per link that is connected) and is created and destroyed at the leads. When these creation/destruction rate pairs are the same in all leads the system obeys detailed balance (aka is an equilibrium problem), when at least one of these rate pairs is different in one lead the system is driven (eg. inject at one end and eject at the other).

The key result of the paper is that generically in this setting a driven system (leads at different temperatures so the rate pairs not all equal) can be mapped to an equilibrium system (leads all at the same temperature) with the (i) same state space and (ii) same connectivity between states, including leads. This is a non-trivial result. For integrable systems (eg. SEP chain with boundary injection/ejection) the mapping can be shown explicitly. The logic is very simple and easy to follow - below I give my perspective on it and ask a couple of questions. The paper, maybe with some of my questions addressed, definitively should be published in SciPost.

The logic of how one arrives to the result is straightforward and that is one thing I like here. I explain if from the perspective I see it most clearly, see refs.[1-3]. What is exploited in Sec.4 is what is sometimes called a "gauge transformation" in Markov chains (the most used one is the so-called Doob transform). This is borrowed from terminology in matrix product states (MPS) which represent hidden Markov chains (and thus also explicit Markov chains).

The probability of a particular trajectory is a product of operators (e.g. exponential of the escape rate during a waiting time, times the probability for a particular jump, and so on). The probability for all trajectories, i.e. the ensemble of trajectories, can thus be collected in an MPS in time. MPS have a gauge invariance, since one can insert identities between every pair of the operators above. It is a gauge symmetry as the operators can be time dependent. The gauge transformation is a change of basis or alternatively a change of generators of the trajectory. Only certain choices of gauge give stochastic generators.

Equation 4.11 is one such gauge transformation *in the long-time limit* where the transformation becomes time-independent. It differs crucially with the Doob transforms used e.g. in large deviation theory in that it is not a point transformation (which trivially leaves the allowed transitions invariant while modifying their rates). Nevertheless, the authors find transformation operators which while not diagonal (as in Doob) still maintain the transition network (thus mapping the problem into itself with different rates). The transformation maps one stochastic generator to a different stochastic generator which an equivalent ensemble of trajectories.

So these are my questions/comments:

- The full equivalence of trajectories at all times (not just long times) requires the time-dependent form of the transformation. Could the authors please comment on this.

- Specifically, the initial probability in the original problem (generated by H) gets mapped to a different probability in the transformed problem (generated by H_eq) by an unpaired gauge operator P. How do these probabilities relate to each other? Furthermore, for the overall probability over trajectories to be well behaved the usual condition is that the transformation reverts eventually to the identity at the final time, see e.g. [3]. Does not the same need to occur here?

- If one is interested in steady state dynamics, the initial state becomes irrelevant. However, in light of the intriguing comment by the authors towards the end about the free-energy of H_eq problem being the non-equilibrium free-energy of the driven problem there is a related consideration. To obtain the probability over states one needs to sandwich (using the MPS language) the train of operators with a configuration vector. But again there is another unpaired transformation operator that maps one basis to the other. I think one then needs to explain more clearly how the steady state probabilities in one problem map to the steady state probabilities in the other. I do not think it is just p(x) = p_eq(x), and thus it is not totally obvious to me what the statement about the free-energy precisely means. Can the authors please comment?

I apologise if some or all of these are already answered in the MS, but in that case I would suggest making those statements more prominent and explicit.

[1] Haegeman J, Cirac J I, Osborne T J and Verstraete F 2013 Phys. Rev. B 88 085118
[2] Chetrite R and Gupta S 2011 J. Stat. Phys. 143 543
[3] Chetrite R and Touchette H 2015 Ann. Henri Poincaré 16 2005

---

## Round 2 · Referee Report · Anonymous · 2020-8-8

Strengths

Establishes a novel mapping between transport and "equilbrium" models for a large familiy of stochastic processes.

Report

The authors show that are fairly wide class of classical stochastic
processes that describe transport are related to equilibrium
models. The class considered contains much studied interacting
many-particle models like the symmetric exclusion process.

It is well known that stochastic processes on lattices can be
represented as imaginary time Schroedinger equations for certain
lattice spin "Hamiltonians". In this context the authors may want to
cite an old work by Alacraz et al Annals Phys. 230 (1994) 250-302.
The authors consider a number of such "Hamiltonians" studied in
transport settings transport and show that they are related by
similarity transformations to "Hamiltonians" that look like they
describe equilibrium situations. I think this work is very interesting
as it generalizes results in the literature to a very large class of
stochastic processes, and I therefore recommend publication. However,
I have a number of questions and comments the authors should consider.

(1) In case of the integrable symmetric exclusion process the similarity
transformation has been explicitly constructed by one of the authors
in a recent work. While this is mentioned above (4.23) it is left unclear
what parts of the following discussion is new, and what is quoted from
said earlier work. I think the authors should be more explicit about
this. As far as I can tell (4.31) is old, but (4.36) is new.

(2) The authors may want to add a reference to the work by Alcaraz et
al mentioned above to establishing the isospectrality of
"Hamiltonians" for stochastic processes differing by operators that
are upper triangular in the right basis, as is the case for (4.21) and
(4.22).

(3) My main question has to do with the authors' statement that their
class of transport models are "in fact hidden equilibrium models". My
understanding is that the authors have established that the evolution
operators of the processes they study can be mapped, by means of a
similarity transformation, to "Hamiltonians" that look like they
describe equilibrium situations. I am confused about the authors
statement quoted above because these "Hamiltonians" are generally not stochastic
and therefore cannot be thought of as generators of stochastic
processes. The steady state of the original stochastic process can of
course by construction be obtained from the "ground state" of the
transformed "Hamiltonian" (now viewed as a quantum spin chain at zero
temperature), but this looks like a purely technical observation to me
(in the sense that expectation values in the quantum model will not
generally be related to averages in the original stochastic
process). I think it would be very helpful if the authors expressed
more precisely in which sense their class of transport models are "in
fact hidden equilibrium models".

---

## Round 3 · List of Changes

We thank the referees for their interesting reports. We modified our manuscript accordingly. Please find the changes below.
%%%%%%%%%%%%%%%%%%%%%%%%%%%%%%%%%%%%
Referee 1:
Questions:
(1) The full equivalence of trajectories at all times (not just long times) requires the time-dependent form of the transformation. Could the authors please comment on this.
(2) Specifically, the initial probability in the original problem (generated by H) gets mapped to a different probability in the transformed problem (generated by H_eq) by an unpaired gauge operator P. How do these probabilities relate to each other? Furthermore, for the overall probability over trajectories to be well behaved the usual condition is that the transformation reverts eventually to the identity at the final time, see e.g. [3]. Does not the same need to occur here?
(3) If one is interested in steady state dynamics, the initial state becomes irrelevant. However, in light of the intriguing comment by the authors towards the end about the free-energy of H_eq problem being the non-equilibrium free-energy of the driven problem there is a related consideration. To obtain the probability over states one needs to sandwich (using the MPS language) the train of operators with a configuration vector. But again there is another unpaired transformation operator that maps one basis to the other. I think one then needs to explain more clearly how the steady state probabilities in one problem map to the steady state probabilities in the other. I do not think it is just p(x) = p_eq(x), and thus it is not totally obvious to me what the statement about the free-energy precisely means. Can the authors please comment?
I apologise if some or all of these are already answered in the MS, but in that case I would suggest making those statements more prominent and explicit.
[1] Haegeman J, Cirac J I, Osborne T J and Verstraete F 2013 Phys. Rev. B 88 085118
[2] Chetrite R and Gupta S 2011 J. Stat. Phys. 143 543
[3] Chetrite R and Touchette H 2015 Ann. Henri Poincaré 16 2005
Answers/modifications:
(1) The mapping in eq. (25) of the revised version is a similarity transformation
between the (transposed) generators of the equilibrium process and of the
non-equilibrium process. As it is now explained in the paragraph
following eq. (15), the mapping provides then the possibility, AT ALL TIMES,
of expressing expectations of the non-equilibrium process via expectations of a
dual absorbing process. Even though the use of the dual process becomes particularly
convenient in the long-time limit where the dual system will be void,
the fact remains that any process in the original system is mappable
to a process either with absorbing baths or with equilibrium baths.
This is interesting as a matter of principle.
We did not investigate further the relation between the paths-space
measures of equilibrium process and of the non-equilibrium process for a given
fixed time.
(2) The relation between the probability distribution of the original process
and the probability distribution of the transformed process is now explicit
in the text that has been added after eq. (26). See also the comment at the end of section 4.
(3) For the mapping between the steady state probabilities in one problem and the steady state probabilities in the other, see the answers above. For the statement about free energies:
this indeed deserves further investigation. While the statement is clear (being the
equilibrium free energy well defined), computing the free energy of the
non-equilibrium system via the mapping to equilibrium requires non trivial work.
This could possibly be done in the integrable case of the open symmetric exclusion process
and should reproduce the density large deviation functionals that has been found
in the macroscopic fluctuation theory of Bertini et al. We plan to reconsider this
problem in a future paper. In any case, for the use of the reader we have added the three references dealing with MPS language.
%%%%%%%%%%%%%%%%%%%%%%%%%%%%%%%%%%%%
Referee 2:
Questions:
(1) In case of the integrable symmetric exclusion process the similarity
transformation has been explicitly constructed by one of the authors
in a recent work. While this is mentioned above (4.23) it is left unclear
what parts of the following discussion is new, and what is quoted from
said earlier work. I think the authors should be more explicit about
this. As far as I can tell (4.31) is old, but (4.36) is new.
(2) The authors may want to add a reference to the work by Alcaraz et
al mentioned above to establishing the isospectrality of
"Hamiltonians" for stochastic processes differing by operators that
are upper triangular in the right basis, as is the case for (4.21) and
(4.22).
(3) My main question has to do with the authors' statement that their
class of transport models are "in fact hidden equilibrium models". My
understanding is that the authors have established that the evolution
operators of the processes they study can be mapped, by means of a
similarity transformation, to "Hamiltonians" that look like they
describe equilibrium situations. I am confused about the authors
statement quoted above because these "Hamiltonians" are generally not stochastic
and therefore cannot be thought of as generators of stochastic
processes. The steady state of the original stochastic process can of
course by construction be obtained from the "ground state" of the
transformed "Hamiltonian" (now viewed as a quantum spin chain at zero
temperature), but this looks like a purely technical observation to me
(in the sense that expectation values in the quantum model will not
generally be related to averages in the original stochastic
process). I think it would be very helpful if the authors expressed
more precisely in which sense their class of transport models are "in
fact hidden equilibrium models".
Answers/modifications:
(1) We have extended the discussion about the relation to the work
of one of the authors and the current article just above (37).
(2) We now refer to the work of Alcaraz in the text preceding eq. (21).
(3) The two Hamiltonian appearing in eq. (25) are both stochastic,
H is the evolution operator of the non-equilibrium process and H_eq
is the evolution operator of the equilibrium process. We added
some text before eq. (27) to make explicit the relation between
the probability distributions of the two processes.
See also the comment at the end of section 4, where it is discussed the
situation in which the initial state is not a probability measure.
%%%%%%%%%%%%%%%%%%%%%%%%%%%%%%%%%%%%
Referee 3:
Questions:
(1) A general notion of duality is mentioned but never introduced in a general setting, apart from stating that it is a well-known duality in probability theory. Also, it is not clear how the existence of a symmetry group for the generator interplays with such duality. Is there a general way to introduce group-based diffusive models and duality independently of the specific cases analyzed?
(2) The technical knowledge displayed of the rich mathematics of interacting particle models is impressive, but at times it looks like a piece of virtuosism (for example: what is the actual need for eq. 4.9? Also: in Sec: 4.3 it is mentioned that, quite intriguingly, integrability allows to solve for W whenever Q_0 is easier to diagonalize than H_0: but it appears that in the following treatment an expression for W is found without diagonalizing H_0)
(3) In the introduction it is mentioned that there exist a time-reversed mapping between \rho_2 and \rho_1 even when the system is driven out of equilibrium. But how is this special of equilibrium systems? Given a MC propagator e^tH one can always define the inverse map e^-tH. Detailed balance has more to do with correlation functions than with probabilities.
(4) It is mentioned in passing that the new detailed balanced dynamics has absorbing sites. But then, the stationary state of the new equilibrium dynamics will be different from the stationary state of the original dynamics. So, while there is a mapping between generators, it appears that there is no mapping between the time-evolved probabilities, and that stationary states map to specific transient states. Maybe the Authors should explain better what exactly the mapping consists of.
Answers/modifications:
(1) We added a paragraph after eq. (15) where we show how our formulation is essentially equivalent
to the notion of duality. We pointed to references discussing the relation between duality
and symmetries, as well as a constructive approach to group-based models with duality. However the referee is right that a complete theory of duality
is still missing.
(2) For clarity we pointed to the formal aspect of formula (4.9). Further, we reformulated the sentence in section 4.3 "Q_0 is easier to diagonalise" in order to clarify that we do not need the eigenvectors of the unperturbed system but the eigenvalues to obtain the similarity transformation W.
(3) Unfortunately e^{-tH} is not a stochastic process.
(4) The precise implication of the mapping is now detailed for the probability vector.
See the added paragraph after eq. (27).
%%%%%%%%%%%%%%%%%%%%%%%%%%%%%%%%%%%%
%%%%%%%%%%%%%%%%%%%%%%%%%%%%%%%%%%%%
Referee 1:
Questions:
(1) The full equivalence of trajectories at all times (not just long times) requires the time-dependent form of the transformation. Could the authors please comment on this.
(2) Specifically, the initial probability in the original problem (generated by H) gets mapped to a different probability in the transformed problem (generated by H_eq) by an unpaired gauge operator P. How do these probabilities relate to each other? Furthermore, for the overall probability over trajectories to be well behaved the usual condition is that the transformation reverts eventually to the identity at the final time, see e.g. [3]. Does not the same need to occur here?
(3) If one is interested in steady state dynamics, the initial state becomes irrelevant. However, in light of the intriguing comment by the authors towards the end about the free-energy of H_eq problem being the non-equilibrium free-energy of the driven problem there is a related consideration. To obtain the probability over states one needs to sandwich (using the MPS language) the train of operators with a configuration vector. But again there is another unpaired transformation operator that maps one basis to the other. I think one then needs to explain more clearly how the steady state probabilities in one problem map to the steady state probabilities in the other. I do not think it is just p(x) = p_eq(x), and thus it is not totally obvious to me what the statement about the free-energy precisely means. Can the authors please comment?
I apologise if some or all of these are already answered in the MS, but in that case I would suggest making those statements more prominent and explicit.
[1] Haegeman J, Cirac J I, Osborne T J and Verstraete F 2013 Phys. Rev. B 88 085118
[2] Chetrite R and Gupta S 2011 J. Stat. Phys. 143 543
[3] Chetrite R and Touchette H 2015 Ann. Henri Poincaré 16 2005
Answers/modifications:
(1) The mapping in eq. (25) of the revised version is a similarity transformation
between the (transposed) generators of the equilibrium process and of the
non-equilibrium process. As it is now explained in the paragraph
following eq. (15), the mapping provides then the possibility, AT ALL TIMES,
of expressing expectations of the non-equilibrium process via expectations of a
dual absorbing process. Even though the use of the dual process becomes particularly
convenient in the long-time limit where the dual system will be void,
the fact remains that any process in the original system is mappable
to a process either with absorbing baths or with equilibrium baths.
This is interesting as a matter of principle.
We did not investigate further the relation between the paths-space
measures of equilibrium process and of the non-equilibrium process for a given
fixed time.
(2) The relation between the probability distribution of the original process
and the probability distribution of the transformed process is now explicit
in the text that has been added after eq. (26). See also the comment at the end of section 4.
(3) For the mapping between the steady state probabilities in one problem and the steady state probabilities in the other, see the answers above. For the statement about free energies:
this indeed deserves further investigation. While the statement is clear (being the
equilibrium free energy well defined), computing the free energy of the
non-equilibrium system via the mapping to equilibrium requires non trivial work.
This could possibly be done in the integrable case of the open symmetric exclusion process
and should reproduce the density large deviation functionals that has been found
in the macroscopic fluctuation theory of Bertini et al. We plan to reconsider this
problem in a future paper. In any case, for the use of the reader we have added the three references dealing with MPS language.
%%%%%%%%%%%%%%%%%%%%%%%%%%%%%%%%%%%%
Referee 2:
Questions:
(1) In case of the integrable symmetric exclusion process the similarity
transformation has been explicitly constructed by one of the authors
in a recent work. While this is mentioned above (4.23) it is left unclear
what parts of the following discussion is new, and what is quoted from
said earlier work. I think the authors should be more explicit about
this. As far as I can tell (4.31) is old, but (4.36) is new.
(2) The authors may want to add a reference to the work by Alcaraz et
al mentioned above to establishing the isospectrality of
"Hamiltonians" for stochastic processes differing by operators that
are upper triangular in the right basis, as is the case for (4.21) and
(4.22).
(3) My main question has to do with the authors' statement that their
class of transport models are "in fact hidden equilibrium models". My
understanding is that the authors have established that the evolution
operators of the processes they study can be mapped, by means of a
similarity transformation, to "Hamiltonians" that look like they
describe equilibrium situations. I am confused about the authors
statement quoted above because these "Hamiltonians" are generally not stochastic
and therefore cannot be thought of as generators of stochastic
processes. The steady state of the original stochastic process can of
course by construction be obtained from the "ground state" of the
transformed "Hamiltonian" (now viewed as a quantum spin chain at zero
temperature), but this looks like a purely technical observation to me
(in the sense that expectation values in the quantum model will not
generally be related to averages in the original stochastic
process). I think it would be very helpful if the authors expressed
more precisely in which sense their class of transport models are "in
fact hidden equilibrium models".
Answers/modifications:
(1) We have extended the discussion about the relation to the work
of one of the authors and the current article just above (37).
(2) We now refer to the work of Alcaraz in the text preceding eq. (21).
(3) The two Hamiltonian appearing in eq. (25) are both stochastic,
H is the evolution operator of the non-equilibrium process and H_eq
is the evolution operator of the equilibrium process. We added
some text before eq. (27) to make explicit the relation between
the probability distributions of the two processes.
See also the comment at the end of section 4, where it is discussed the
situation in which the initial state is not a probability measure.
%%%%%%%%%%%%%%%%%%%%%%%%%%%%%%%%%%%%
Referee 3:
Questions:
(1) A general notion of duality is mentioned but never introduced in a general setting, apart from stating that it is a well-known duality in probability theory. Also, it is not clear how the existence of a symmetry group for the generator interplays with such duality. Is there a general way to introduce group-based diffusive models and duality independently of the specific cases analyzed?
(2) The technical knowledge displayed of the rich mathematics of interacting particle models is impressive, but at times it looks like a piece of virtuosism (for example: what is the actual need for eq. 4.9? Also: in Sec: 4.3 it is mentioned that, quite intriguingly, integrability allows to solve for W whenever Q_0 is easier to diagonalize than H_0: but it appears that in the following treatment an expression for W is found without diagonalizing H_0)
(3) In the introduction it is mentioned that there exist a time-reversed mapping between \rho_2 and \rho_1 even when the system is driven out of equilibrium. But how is this special of equilibrium systems? Given a MC propagator e^tH one can always define the inverse map e^-tH. Detailed balance has more to do with correlation functions than with probabilities.
(4) It is mentioned in passing that the new detailed balanced dynamics has absorbing sites. But then, the stationary state of the new equilibrium dynamics will be different from the stationary state of the original dynamics. So, while there is a mapping between generators, it appears that there is no mapping between the time-evolved probabilities, and that stationary states map to specific transient states. Maybe the Authors should explain better what exactly the mapping consists of.
Answers/modifications:
(1) We added a paragraph after eq. (15) where we show how our formulation is essentially equivalent
to the notion of duality. We pointed to references discussing the relation between duality
and symmetries, as well as a constructive approach to group-based models with duality. However the referee is right that a complete theory of duality
is still missing.
(2) For clarity we pointed to the formal aspect of formula (4.9). Further, we reformulated the sentence in section 4.3 "Q_0 is easier to diagonalise" in order to clarify that we do not need the eigenvectors of the unperturbed system but the eigenvalues to obtain the similarity transformation W.
(3) Unfortunately e^{-tH} is not a stochastic process.
(4) The precise implication of the mapping is now detailed for the probability vector.
See the added paragraph after eq. (27).
%%%%%%%%%%%%%%%%%%%%%%%%%%%%%%%%%%%%

---

## Editorial Decision

published